



# Can we assess the value meteorological ensembles add to dispersion modelling using hypothetical releases?

Susan J. Leadbetter[1], Andrew R. Jones[1], and Matthew C. Hort[1]

[1]Met Office, Exeter, EX1 3PB, UK

**Correspondence:** Susan Leadbetter (susan.leadbetter@metoffice.gov.uk)

**Abstract.** Atmospheric dispersion model output is frequently used to provide advice to decision makers, for example, about the likely location of volcanic ash erupted from a volcano or the location of deposits of radioactive material released during a nuclear accident. Increasingly scientists and decision makers are requesting information on the uncertainty of these dispersion model predictions. One source of uncertainty is in the meteorology used to drive the dispersion model and in this study ensemble meteorology from the Met Office ensemble prediction system is used to provide meteorological uncertainty to dispersion model predictions. Two hypothetical scenarios, one volcanological and one radiological, are repeated every 12 hours over a period of 4 months. The scenarios are simulated using ensemble meteorology and deterministic forecast meteorology and compared to output from simulations using analysis meteorology using the Brier skill score. Adopting the practice commonly used in evaluating numerical weather prediction models (NWP) where observations are sparse or non-existent we consider output from simulations using analysis NWP data to be truth. The results show that on average the ensemble simulations perform better than the deterministic simulations although not all individual ensemble simulations outperform their deterministic counterpart. The results also show that greater skill scores are achieved by the ensemble simulation for later time steps rather than earlier time steps and at those later time steps for deposition than for air concentration. For the volcanic ash scenarios it is shown that the performance of the ensemble at one flight level can be different to that at a different flight level, e.g. a negative skill score might be obtained for FL350-550 and a positive skill score for FL200-350. This study does not take into account any source term uncertainty but it does take the first steps towards demonstrating the value of ensemble dispersion model predictions.




# 1 Introduction

The release of natural and man-made contaminants into the atmosphere can pose a hazard to human health, animal and plant health and infrastructure. The dispersion and deposition of these contaminants is routinely simulated using atmospheric disper-
25 sion models. Output from these simulations is frequently used to provide advice to decision makers and health professionals on the possible level of exposure. For example, dispersion models are used to forecast the transport of volcanic ash following an eruption and the forecasts are used by the aviation industry to reduce the risk of damage to aircraft. Dispersion models are also used to provide estimates of areas where radionuclide concentrations will result in health intervention levels being exceeded following a nuclear accident.

The accuracy of the simulations produced by atmospheric dispersion models are dependent not only on the numerical approximations and physical parameterisations within the dispersion model, but also on the inputs to the model, the meteorological and source term information. Meteorological information is typically provided as four-dimensional meteorological fields from numerical weather prediction (NWP) models although some dispersion models can also use meteorological observations taken from a single observing station. Source term information typically comprises of information about the material released such
as the quantity of material released, the height and timing of the release and the composition of the release. Information about the material being released may be obtained from emission models or observations depending on the type of release.

The meteorological and source term inputs and the physical and numerical parameterisations in the dispersion model all contain uncertainties. General discussions on the types of uncertainties in dispersion models and their relative importance are given by Rao (2005) and Leadbetter et al. (2020). At the Met Office, two approaches are currently taken to evaluate these
40 uncertainties. First, one or two additional simulations (called scenarios) are carried out where the source term information is perturbed (Beckett et al., 2020; Millington et al., 2019). Second, uncertainties in the source term and the meteorology are evaluated individually and qualitatively by including a general statement of the uncertainties in a text or verbal description accompanying the model output. However, in recent years there has been a growth in the scientific understanding of the uncertainties and the ability to represent them in a model framework. In addition there is an increasing expectation from both
scientific advisors and decision makers to provide a quantitative evaluation of these uncertainties.

There are a number of challenges to providing quantitative estimates of uncertainty in dispersion model predictions. First, most dispersion incidents require a rapid response so there is limited time to carry out uncertainty evaluations and comminicate them. Second, there are a relatively small number of dispersion incidents for which sufficient observations exist to assess the performance of the uncertainty estimates. These challenges have resulted in relatively few studies of dispersion uncertainty.
Netherthless, several studies have assessed the sensitivity of the output to dispersion model parameters and input variables. Statistical emulators have been applied to model predictions for the Fukushima accident (Girard et al., 2016) and the eruption of Eyjafjallajökull in 2010 (Harvey et al., 2018). They showed dispersion predictions for the Fukushima accident are sensitive to wind speed, wind direction, precipitation and emission rate and dispersion predictions for the part of the eruption of Eyjafjallajökull around the 14 May are more senstive to initial plume height, mass eruption rate, free tropospheric turbulence and the
threshold precipitation above which wet deposition occurs. Due to the computational expense of running a statistical emulator





these two studies are limited to just two dispersion events and thus to two meteorological scenarios. In more idealised studies, not focussed on a single event, Haywood (2008) demonstrated the sensitivity of a surface release to wind direction and speed.

A comprehensive study of all three sources of dispersion model uncertainty is too large a topic for a single paper so here we focus on the meteorological uncertainty. Galmarini et al. (2004) describes three methods for providing meteorological uncertainty information to a dispersion model; perturbations to the initial dispersion conditions, a suite of different NWP models, and a single NWP with perturbations to the initial conditions and model physics. The first of these approaches could be used to explore uncertainty in the source information or the meteorology. For example, Draxler (2003) perturbed the location of the source horizontally and vertically to explore the sensitivity of the dispersion model HYSPLIT to small perturbations in meteorology. When modelling the Across North America Tracer Experiment (ANATEX) with this method he was able to account for around 45% of the variance in the measurement data. The second approach has been used in a few post event analysis projects (e.g., Galmarini et al., 2010; Draxler et al., 2015) and in general these studies demonstrate that an ensemble of dispersion predictions outperforms a single dispersion prediction when compared to observations. However, for a single institute running multiple NWP models would be prohibitively expensive both in terms of human and computational resource. In addition Galmarini et al. (2010) demonstrated that for the European Tracer Experiment (ETEX) the performance of an ensemble prediction system based on a single NWP was comparable to the performace of multi-NWP model ensemble. The third approach, using a single NWP with perturbations to the intial conditions and/or the model physics has also been used to model the Fukushima accident (e.g., Korsakissok et al., 2020). The Met Office has recently added the ability to run an ensemble dispersion model operationally using this third approach and it is this approach that we focus on here.

Developers of numerical weather prediction (NWP) models represent the uncertainty in the atmospheric state and its evolution by running multiple model integrations where each model integration starts from a perturbed initial model state and uses perturbed model physics. These are known as 'ensemble' models and were first used for weather forecasting in the 1990s. They can also be used to provide information about the meteorological uncertainty to dispersion models. Meteorological ensembles were first used with dispersion models in the late 1990's/early 2000's when output from a dispersion model (SNAP) driven by ensemble meteorology from ECMWF (European Centre for Medium-range Weather Forecasting) was compared to observations taken as part of the ETEX experiment (Straume et al., 1998; Straume, 2001). The results of this study demonstrated that for the meteorological conditions during ETEX the ensemble mean performed better than the control. However, (Straume, 2001) also noted that at that time there was insufficient computational power to run ensemble dispersion models routinely.

Computational power is no longer a barrier to running ensemble dispersion models and more recently ensemble meteorology has been used with dispersion models to study the Fukushima accident and the eruption of several volcanoes (Eyjafjallajökull in April 2010, Grimsvötn in May 2011, Kelut in February 2014 and Rinjani in November 2015). The Fukushima accident has been simulated using ensemble meteorology from ECMWF, source terms estimated after the event using reverse or inverse techniques and a number of different dispersion models (Périllat et al., 2019; Korsakissok et al., 2020). However, these simulations were not compared to single meteorological model/single source-term simulations so do not provide an indication of whether ensemble meteorology outperforms deterministic meteorology for this case study.



The Kelut eruption in 2014 was simulated by Dare et al. (2016) using ensemble meteorology from the Australian Community Climate and Earth System Simulator (ACCESS) within the dispersion model HYSPLIT. They showed that the ensemble dispersion simulation compared better, qualitatively, to the satellite observations than the deterministic dispersion simulation. They also showed that the ensemble output could be used to highlight the positional uncertainty of the region of maximum concentration of volcanic ash. Zidikheri et al. (2018) also demonstrated that the ACCESS-GE ensemble performed better than

the ACCESS-R (deterministic) regional model when compared to observations of volcanic ash from the eruptions of Kelut and Rinjani in 2014 and 2015 respectively. They used the Brier skill score to show that a 24-member ensemble performed better than the regional model for both eruptions. However, although the lagged ensemble outperformed the most recent ensemble for Kelut the most recent ensemble performed better for Rinjani.

These studies suggest that for those events that have been examined dispersion ensembles outperform dispersion models run

using a single meteorological model. However, they are focused on just a few events covering relatively short periods of time and with the exception of Fukushima they are all volcanic releases extending several kilometres into the atmosphere and so cannot be considered to be representative of a release within the boundary layer.

In order to assess the value of using meteorological ensembles with dispersion models we need to assess the performance of the ensemble over a large range of meteorological conditions. In addition, verification of ensembles requires larger data sets

than the verification of deterministic output due to the extra probabilistic 'dimension' (Wilks, 2019). To increase the size of our data set we borrow a method regularly used to verify meteorological models and verify our ensemble dispersion output against dispersion output produced using analysis meteorological data (see for example Ebert et al. (2013)). Meteorological modellers use a number of methods to verify their models and there are advantages and disadvantages of each method. One method is to use analysis data for verification of variables that are not easily observed and to produce gridded fields of variables that are only

measured at a few sparsely located observations sites. This reduces the errors that can results from comparing model grid box averages with point observations (Haiden et al., 2012). However, the processing necessary to produce the analysis data using a combination of observations and knowledge of atmospheric processes potentially introduces some additional uncertainty (Bowler et al., 2015).

From a dispersion modelling perspective simulations using analysis meteorology are dispersion simulations using the best

estimate of the meteorological conditions. The simulations don't take into account uncertainty in the source term or uncertainty in the dispersion model itself. There are a number of advantages to this approach. First, we can assess the performance of the dispersion ensemble over a large range of meteorological conditions. Second, we remove any source term uncertainty allowing us to independently assess the meteorological uncertainty alone. Third, we can examine case studies within and outside the boundary layer to understand the value of ensembles for releases at different heights accepting that NWP ensembles

including ensembles may be configured to perform better for certain variables and some parts of the atmosphere. Despite these advantages, we also need to be aware of the disadvantages of verifying against analysis meteorology. The analyis meteorology is constructed using a model so still contains uncertainties and during and following an incident dispersion models would be compared to observations not analysis model data.





## 2 Method

To explore whether using ensemble meteorology rather than a single meteorological model can add value to a dispersion prediction two scenarios were constructed; one focused on a boundary layer radiological release and the other on a volcanic eruption releasing ash into the troposphere and lower stratosphere.

The first scenario considered a radiological release. Accidental releases from nuclear power plants involve many different radionuclides but as the aim of this study was to explore meteorological uncertainty rather than source term uncertainty a

130 hypothetical release of 1PBq caesium-137 over six hours at an elevation of 50m was used. It was assumed that the caesium-137 was carried on particles with a diameter less than $1\mu m$ and had a decay rate of 30 years. To sample a range of different meteorological scenarios, the release was simulated from 12 different locations with different topographical situations, and coastal and non-coastal environments across Europe (see Figure 1(a)). Note that these locations are not known locations of nuclear facilities, they were just chosen due their topographic and coastal situation. For this scenario, total integrated air

concentrations and total deposition for the full 48 hours were output. All quantities were output on a grid with a resolution of $0.141^o$ longitude by $0.0943^o$ latitude (approximately 10km by 10km at mid-latitudes).

The second scenario considered a hypothetical eruption of two volcanoes in Iceland; an eruption of Hekla with an eruption height of 12km and an eruption of Oraefajokull with an eruption height of 25km (see Figure 1(b)). The runs were set up in the same manner as the operational runs at the London Volcanic Ash Advisory Centre (VAAC) (see Beckett et al. (2020) for more

details). Mass eruption rates were computed using a relationship between eruption height and eruption rate proposed by Mastin et al. (2009) and assuming that only 5% of the ash is small enough to be transported over long distances in the atmosphere. This results in release rates of 8.787e12 g/hr and 1.131e14 g/hr for Hekla and Oraefajokull respectively. The particle size distribution is based on the eruption of Mount Redoubt in 1990 and particles have a density of $2300kg/m^3$. The eruptions were assumed to last for 24 hours and transport and deposition of the emitted ash was modelled for 24 hours. Airborne ash

concentrations were output and processed following the same procedure as the London VAAC. First, ash concentrations were averaged over thin layers 25 FL deep ( 800m), where flight levels (FL) represent aircraft altitude at standarad air pressure and are approximately expressed as hundreds of feet. Then the thin layers were combined into the three thick layers (FL000-200, FL200-350, FL350-550) by taking the maximum ash concentration within the thin layers, which make up a thick layer, and applying it to the entire depth (Webster et al., 2012). In addition accumulated deposits and 3-hourly vertically integrated ash

concentrations (hereafter referred to as ash column load) were also output. All quantities were output on a grid with a resolution of $0.314^o$ latitude by $0.179^o$ longitude (approximately 20km by 20km at mid-latitudes).

To explore a range of meteorological conditions both scenarios were repeated every 12 hours over a period of around 4 months (03/11/2018-28/02/2019 for the radiological scenario and 01/12/2018-31/03/2019 for the volcanic eruption scenario) with each simulation being run on single NWP forecast. Technical issues resulted in the loss of some of the simulations on

the 18 and 19 January. Therefore a total of 232 simulations of the radiological scenario and 240 simulations of the volcanic eruption scenario were carried out. In both scenarios the release start times are 6 hours after the meteorological forecast data initialisation time.





### 2.0.1 Overview of NAME

Dispersion modelling was carried out using NAME (Numerical Atmospheric-dispersion Modelling Environment), the UK Met
Office's Lagrangian particle dispersion model. NAME is used to model the atmospheric transport and dispersion of a range
of gases and particles (Jones et al., 2007). It is the operational model of the London VAAC (Volcanic Ash Advisory Centre)
responsible for forecasting the dispersion of ash in the northeast Atlantic and over the UK, Ireland and Scandinavia and it
is also the operational model of RSMC Exeter (Regional Specialist Meteorological Centre) responsible for forecasting the
dispersion of radioactive material in Europe and Africa. In NAME, large numbers of computational particles are released
into the model atmosphere with each computational particle representing a proportion of the mass of the material (gases or
particles) being modelled. Computational particles are advected within the model atmosphere by three-dimensional winds from
numerical weather prediction models and turbulent dispersion is simulated by random walk techniques. Mass is removed from
the model atmosphere by wet and dry deposition as well as by gravitational settling for volcanic ash and by radioactive decay
for caesium-137.

### 2.0.2 Meteorological Data

Meteorological datasets for this study are provided by three different model configurations of the Met Office's Unified Model
(Walters et al., 2019). The ensemble meteorological data is provided by the global configuration of the Met Office Global
and Regional Ensemble Prediction System (MOGREPS-G) which is an ensemble forecasting system developed and run op-
erationally at the Met Office (Tennant and Beare, 2014). It is an 18-member ensemble which runs four times a day at 00, 06,
12 and 18 UTC. In the global configuration it runs at a resolution of $0.28125^o$ latitude by $0.1875^o$ longitude (approximately
20km by 20km at mid-latitudes) with 70 vertical levels extending from the surface up to 80km (in these simulations only the
first 59 levels extending up to 30km are used). At the time this work was carried out initial conditions were obtained from
the global deterministic 4D-Var data assimilation system with perturbations generated using an Ensemble Transform Kalman
Filter (ETKF) approach. Model perturbations followed a stochastic physics approach where the tendency of model parameters
such as temperature were perturbed.

The ensemble dispersion simulations are compared to dispersion runs carried out using the global deterministic configuration
of the Met Office Unified Model (hereafter referred to as the deterministic forecast). The global deterministic configuration of
the Unified Model (Walters et al., 2019) has a resolution of $0.140625^o$ latitude by $0.09375^o$ longitude (approximately 10km by
10km at mid-latitudes) and has the same vertical level set as MOGREPS-G. It runs four times a day, two 168-hour forecasts at
00 and 12 UTC and two update forecasts of 69 hours at 06 and 18 UTC. In this study the focus is on the two forecasts at 00
and 12 UTC. An analysis meteorological data set is constructed by stitching together the first 6 hours of each 6-hourly forecast
and the dispersion simulations are repeated using this data set. This means that for the first 6 hours of meteorological data the
deterministic forecast and the analysis meteorology will be identical. Therefore to avoid giving the deterministic meteorology
a skill advantage the simulated releases are 6-hours after the meteorological forecast data initialisation time.


### 2.0.3 Assessment of Ensemble Skill

There are many ways to assess the performance of an ensemble (Wilks, 2019). Some of these measure attributes such as reliability which measures the degree to which simulated probabilities match observed frequencies, and resolution that measures the ability of the ensemble to distinguish between events with different frequencies, can only be calculated for a (large) set of ensembles. The aim of this study is to evaluate the skill of each individual simulation and to do this the Brier score relative to the analysis (Brier, 1950) is used. The Brier score is commonly used to evaluate meteorological ensembles and is completely analogous to the mean-square-error measure of accuracy used for deterministic forecasts. The Brier score is a measure of the accuracy of a forecast in terms of the probability of the occurrence of an event. For a set of $N$ forecasts it is typically expressed as:

$$BS = \frac{1}{N} \sum_{i=1}^{N} (f_i - a_i)^2 \tag{1}$$

where $f_i$ is the forecast probability of the event occurring for the $ith$ forecast, and $a_i$ is the observation of the event, with $a_i = 0$ if the event does not occur and $a_i = 1$ if it does occur. Here the dispersion prediction using the analysis meteorology is substituted for the observation. The Brier score ranges from 0 to 1 and is negatively oriented so that a perfect forecast is assigned a Brier score of 0. For each scenario an event was considered to have occurred if a threshold concentration (or deposition amount) was exceeded in a single output grid square.

Several thresholds were used for each of the scenarios (see Table 1). The simplification of the release rate for the radiological scenario means that there are no relevant published thresholds. Instead thresholds for the radiological scenario were chosen to reflect typical distances at which observations might be made during a radiological release. Accumulated deposition thresholds were chosen so that the areas where deposition exceeded the thresholds were similar to the areas where the total integrated air concentration exceeded the thresholds. Average (mean) maximum distances at which thresholds are exceeded are calculated by computing the maximum distance at which each threshold is exceeded for every location, every ensemble member and every simulation and then averaging over the ensemble members and simulations.

It can be seen that the average maximum distance from the release location that the highest threshold is exceed is around 100km for all of the release locations (see Figure 2). The average maximum distance that the lowest threshold is exceeded is between approximately 650km and 1000km from the release location for the total integrated air concentration and between approximately 700km and 1300km from the release location for the accumulated deposition. The average maximum distances at which thresholds were exceeded are lowest for the releases from Milan, Italy and highest for the releases from Felixstowe, UK reflecting the different weather patterns that influence these locations. Milan, Italy is surrounded by the Alps on the north and west sides so is sheltered from the westerly and southwesterly prevailing winds. In contrast Felixstowe is in a low-lying region of the UK with no shelter from the prevailing westerly and southwesterly winds and the prevailing winds carry material across the sea where winds speeds are typically higher than over land.

Thresholds for the volcanic ash scenario air concentration values reflect values discussed in the literature. The London VAAC currently uses thresholds of 0.2, 2.0 and 4.0 $mg/m^3$ (Beckett et al., 2020). Prata et al. (2019) notes that thresholds between





2.0 and 10.0 $mg/m^3$ have been discussed in meetings with aviation stakeholders, and studies of engine damage consider damage against a log scale of air concentration (Clarkson and Simpson, 2017). Therefore, the thresholds for air concentration

of volcanic ash used in this study range from 0.2 to 10.0 $mg/m^3$ on a log scale. Threshold concentrations for ash column load assume that layers of volcanic ash are on average $1km$ thick so are chosen to be a factor of 1000 larger than the thresholds for air concentration. Accumulated deposition thresholds were chosen so that the areas where deposition exceeded the thresholds were similar to the areas where the air concentration exceeded the thresholds. Average maximum distances at which thresholds are exceeded for each of the three flight levels for the Hekla scenario are shown in Figure 3. As expected the average maximum

distance at which the thresholds are exceeded increases over time. Three hours after the start of the eruption all of the thresholds are exceeded to a maximum distance of around 250km in all three flight levels. Twenty-four hours after the start of the eruption the average maximum distance over which thresholds are exceeded ranges from approximately 1600km to 2400km. To provide a sense of scale 1600km is the approximate distance from Reykjavik, Iceland to Bern, Switzerland. The smallest distances are for the highest thresholds in FL000-200 and FL350-550 and the largest distances are for the lowest thresholds in FL200-250.

The Brier score is often compared to a reference Brier score to produce a Brier skill score. In meteorology the most commonly used reference is climatology. However, there is no climatology for a single release dispersion event so instead a reference forecast is used. In this case the forecast using the global Unified Model is used as a reference as it also demonstrates whether or not the ensemble forecast outperforms the deterministic forecast. This isn't perfect because the deterministic and analysis meteorology have the same resolution while the ensemble meteorology has a coarser resolution. However, all three

meteorological configurations are initialised using the same observations. The Brier skill score is expressed as:

$$BSS = 1 - \frac{BS}{BS_{ref}} \qquad (2)$$

In this case the Brier skill score indicates the level of performance of the ensemble over the deterministic forecast so ensemble forecasts that outperform their deterministic counterparts have positive Brier skill scores and ensemble forecasts that perform worse than their deterministic counterparts have negative Brier skill scores.

## 3  Results

In this section, the Brier skill scores for the ensemble runs are presented. First the average Brier skill scores are presented for each scenario and each release location. Then a few simulations where the skill score was high or low are examined to provide examples of the situations where simulations using ensemble meteorology outperform those using deterministic meteorology and vice versa. It should be noted that the Brier skill scores only provide an assesment of the relative performance of the

ensemble forecast when compared to the deterministic forecast rather than an absolute measure of the performance of the ensemble forecast.





## 3.1 Radiological Scenario

Figure 4 shows the average Brier skill score at each of the 12 locations where a hypothetical radiological release was simulated. Seven threshold values are shown for each of the 48-hour integrated air concentration and the 48-hour accumulated deposition.
Average skill scores are higher for the accumulated deposition than the air concentration and there is more variation in skill score between the different locations for the accumulated air concentration. The highest average skill scores are at Mace Head and the lowest skill scores at the lower threshold values can be found at Milan. However, there is a greater range of skill scores between the thresholds at Inverness and the average skill score for the highest threshold is negative implying that for this threshold, on average, the dispersion runs using deterministic meteorology performed better than the dispersion runs using
ensemble meteorology. For the accumulated air concentration the lowest skill scores were seen for the highest threshold at 8 out of the 12 release locations. This is typically because the highest thresholds are exceeded close to the release location and soon after the release time when the ensemble performs less well than the deterministic simulations (not shown). The same was true for the accumulated deposition although for a different set of 8 release locations.

Although the average Brier skill score at each location is positive implying that on average the ensemble performs better than
265 the deterministic simulation, there are simulations at all locations for which the skill score is negative (Figure 5). The standard Brier skill score can take any value from $-\infty$ to 1 but this means that it is difficult to assess the relative size of negative and positive Brier skill scores on a single plot. Therefore, in Figure 5 the Brier skill score has been adjusted so that negative Brier skill scores lie in the range $[-1,0)$. This adjusted score is defined as:

$$BSS\_adj = \begin{cases} BSS & BSS \geq 0 \\ \frac{BSS}{1-BSS} & BSS < 0 \end{cases} \qquad (3)$$

Brier skill scores are negative for between 10 and 30 percent of the simulations of total accumulated air concentration and between 0.5 and 19 percent of the simulations of accumulated deposition. The highest percentage of negative skill scores are from the simulations of releases at Karlsruhe and Milan for integrated air concentration and deposition respectively and the lowest percentage of negative skill scores are from the simulations of releases at Warsaw for both integrated air concentration and deposition. Generally simulations from release locations with a larger percentage of negative skill scores for air concentration
also have a larger percentage of negative skill scores for deposition.

## 3.2 Volcanic Ash Scenario

For the volcanic ash scenarios Brier skill scores were computed for the air concentration on three flight levels, FL000-200, FL200-350 and FL350-550, the air concentration integrated over the whole depth of the model atmosphere (the ash column load) and the accumulated deposits. Average unadjusted Brier skill scores are shown in Figure 6 and it can be seen that after the
280 first timestep the average scores are greater than zero suggesting that on average the ensemble outperforms the deterministic forecast. Brier skill scores increase with forecast time, and the increase is fastest for the accumulated deposition and slowest





for the highest flight level (FL350-550) for the 12km eruption scenario. Generally, average skill scores are higher for smaller thresholds, although the difference in skill score between the thresholds is smaller than the difference in the skill scores between the forecast times. With exception to the upper two flight levels skill scores are similar for the 12km eruption scenario and the 285    25km eruption scenario. The 12km eruption will only a emit a small amount of material into the highest flight level, FL350-550 because FL350 is typically around 11km above sea level.

Although the average Brier skill scores are generally greater than zero, at each forecast time step there are runs where the Brier skill score is negative suggesting that the deterministic forecast outperforms the ensemble (see Figure 7 which shows individual adjusted Brier scores). There are fewer negative scoring runs at later forecast time steps implying that the ensemble 290    is more likely to perform better than the deterministic at later time steps. This is possibly due to the increase in ensemble spread at later time steps.

### 3.3    Examples of High and Low Brier Skill Score

The Brier skill score is a statistical method of quantifying the performance of an ensemble of predictions against a reference prediction. However, it hides a lot of detail. In this section examples of simulations where the Brier skill score is high and 295    low are examined in more detail. Examples of high and low Brier skill score were selected from the radiological scenario by considering predictions where the adjusted Brier skill score was greater than 0.6 or less than -0.6 for four or more thresholds for a single simulation. The subset of examples in Figure 8 was then randomly selected from the simulations fitting these criteria.

Figure 8(a) shows an example of a high Brier skill score for a prediction of integrated air concentration following a hypothetical release from Milan in northern Italy. In this example the deterministic simulation predicted some transport to the south west 300    and a large amount of transport to the south east broadly following the Po Valley. In contrast the analysis simulation predicted a small amount of transport to the south east and a greater amount of transport to the south west. The ensemble simulation spans both of these predictions and the area where more than 60% of ensemble members exceed the threshold extends a similar distance in the southwesterly and southeasterly direction. This resulted in a Brier skill score of 0.76.

Figure 8(b) shows an example of a low Brier skill score for a prediction of air concentration following a hypothetical release 305    from Kristiansand. In this example the region where the threshold is predicted to be above $200kBqs/m^3$ is very similar in the deterministic and analysis runs, extending due west from the release site. In contrast most ensemble members predict that the threshold will be exceeded in a region slightly further north. At the time of this release a high pressure system was located over Norway and a low pressure system was located to the south of Greenland. This resulted in a high gradient of wind speed at and a change in wind direction close to Kristiansand (see Figure 9(a)). At the start of the release wind speeds at Kristiansand were 310    greater in all the ensemble members than both the deterministic and analysis meteorological data and wind directions were more easterly and less southerly in all the ensemble members than both the deterministic and analysis meteorological data (see Figure 9(b) and (c)). This suggests that the location of the highest gradient in wind speed and change in wind direction were slightly different in all ensemble members resulting in the different predictions of air concentration.

Figure 8(c) shows an example of a high Brier skill score for a prediction of accumulated deposition following a hypothetical 315    release from Kristiansand. The area where the deposits exceed $2kBq/m^2$ is complex because the deposition is dominated by





wet deposition. The area exceeding the threshold predicted by the deterministic simulation covers a large area of the North Sea between the Norwegian coast and the Shetland Islands. However, the area exceeding the threshold in the analysis simulation is mostly limited to a narrow region immediately to the west of the release site. There are three regions where more than 60% of ensemble members are in agreement that the threshold will be exceeded; one in the narrow region immediately to the west

of the release site, one extending from $0 - 4°$E at $59.4°$N and one close to the western most point of the Norwegian coast. Both the ensemble and the deterministic simulation show relatively poor agreement with the analysis simulation but because, in most areas, only a small proportion of the ensemble exceeds the threshold it has a lower Brier score than the deterministic and thus the Brier skill score is positive.

Figure 8(d) shows an example of a low Brier skill score for a prediction of accumulated deposition following a hypothetical

release from Mace Head in Ireland. In this case the area where deposits are predicted to exceed $5kBq/m^2$ is similar in the analysis and deterministic simulations. Although this region closely matches the region where 60% of the ensemble members exceed the same threshold, the good agreement between the analysis and the deterministic simulations coupled with a few ensemble members predicting the threshold will be exceeded further to the north results in a lower Brier score for the deterministic than for the ensemble simulations. This is an example of a case where the ensemble forecast is unable to show an

improvement on the deterministic forecast because the deterministic forecast performs highly. The negative Brier skill score only provides a comparison of the performance of the ensemble relative to the deterministic and does not provide information about the individual performance of the ensemble.

For the volcanic ash scenario Brier skill scores were closer to zero so examples of high and low Brier skill score were selected by considering all simulations where the skill score was greater than 0.5 or less than -0.5 for at least two time steps at

335 a single flight level. Examples were randomly chosen from the simulations meeting these criteria.

Figure 10 shows an example of a simulation where the Brier skill score is positive implying that the ensemble simulation performs better than the single deterministic simulation. The simulation considers a hypothetical volcanic eruption at Hekla volcano with an eruption height of 12km starting at 18:00 UTC on the 21 January 2019. Figure 10 shows regions where the air concentration in FL000-200 exceeds $2mg/m^3$. Volcanic ash is initially transported in a north-easterly direction. Six hours

after the start of the eruption material to the north of Iceland is transported to the north west while newly emitted material is transported to the south east resulting in a bi-directional plume stretching in a north-westerly and south-easterly direction from the volcano.

The Brier skill score for this simulation is positive for all flight levels and all time steps except six hours after the eruption where there are negative and zero skill scores for concentrations exceeding $2mg/m^3$ at heights of FL200-350 and FL000-200

respectively (see Figure 11). The skill score for the lowest flight level gradually increases from six hours after the eruption to 21 hours after the eruption before decreasing slightly. In the middle flight level a similar pattern is observed although in this case the highest skill score occurs 9 hours after the eruption and in the highest flight level there is a slight downward trend in the skill score over time. This demonstrates how ensemble skill, relative to deterministic simulation skill, can vary with height.

What do these Brier skill scores mean for the difference between the ensemble, deterministic and analysis simulations for

the concentration of ash exceeding $2mg/m3$ at FL000-200? Six hours after the start of the eruption there is good agreement





between the deterministic and analysis simulations but there are a few members of the ensemble simulation predicting greater plume spread in an east-west direction. This is consistent with a Brier skill score close to zero. 12 hours after the start of the eruption the deterministic and analysis simulations start to diverge, particularly at the north-western end of the plume and 24 hours after the start of the eruption the analysis simulation predicts that the plume will just reach the coast of Greenland, but the deterministic simulation predicts that the plume will extend approximately five degrees further west. A few ensemble members also predict that the plume will extend several degrees to the west of the coast of Greenland. However, the region where all ensemble members are in agreement that the air concentration will exceed a threshold of $2mg/m^3$ is in good agreement with the region where the analysis simulation exceeds the same threshold. The Brier skill scores for 12, 18 and 24-hours after the start of the eruption are 0.407, 0.502 and 0.484 respectively, indicating that during this time period the ensemble outperforms the deterministic simulation.

Figure 12 shows an example of a volcanic ash simulation where the Brier skill score is negative implying that the deterministic simulation outperforms the ensemble simulation. This simulation considers a hypothetical eruption of Oraefajokull volcano with an eruption height of 25km starting at 06:00 UTC on the 24 December 2018. Material in the lowest flight level (FL000-200) is transported eastwards from Iceland. There is also some southward transport of the ash to the east of Iceland and this increase with each time step so that the region exceeding $5mg/m^3$ extends down the North Sea across the Shetland Islands, the south and west coasts of Norway and much of Denmark.

The Brier skill score for ash concentrations exceeding $5mg/m^3$ in the upper two flight levels, FL200-350 and FL350-550, for this simulation are both positive and change little over the duration of the simulation (see Figure 13). However, at the lowest flight level, FL000-200, the skill score for ash concentrations exceeding $5mg/m^3$ is initially very negative then gradually increases over time becoming positive 21 hours after the eruption.

In the lowest flight level, there is good agreement between the regions where the ash concentration is predicted to exceed $5mg/m^3$ in the deterministic and analysis runs. In addition there is very little spread in the ensemble and the region where 80% of ensemble members predict volcanic ash concentrations to exceed the threshold closely matches the region of threshold exceedance from the analysis simulation. There is a small mismatch in the northern edge of the region where 80% of ensemble members predict concentrations above the threshold and where the analysis and deterministic simulations predict concentrations above the threshold and this results in a negative Brier skill score.

In this section six example simulations have been considered, three where the Brier skill score had a large positive value and three where the skill score had a large negative value. The large positive values can be attributed to simulations where the deterministic simulation performed poorly compared to the analysis and the region where the ensemble predicted the highest probabilities of exceeding a threshold compared well to the analysis simulation. The large negative values occurred in simulations where the deterministic simulation performed well compared to the analysis. In this case the ensembles demonstrated different behaviour, in one case the ensemble predicted transport in a different direction to the analysis, in the second case there was some spread in the ensemble and in the final case there was very little spread in the ensemble.



## 4  Conclusions

This study considers how well a dispersion ensemble constructed using input from a meteorological ensemble model might be expected to perform when compared to a dispersion model using single model deterministic meteorology. Meteorology from the Met Office MOGREPS-G ensemble prediction system is used as input to the NAME dispersion model generating an 18 member dispersion ensemble. The dispersion output is then compared to runs using forecast meteorology from the global deterministic configuration of the Met Office Unified model (referred to as deterministic meteorology). To provide a 'ground 390 truth' an analysis meteorological data set is constructed by stitching together the first 6 hours of each 6-hourly deterministic meteorological forecast.

Dispersion output from two hypothetical scenarios is explored; the first scenario is a near-surface release of radioactive material and the second scenario is a volcanic eruption in Iceland. Simulations of both scenarios are repeated over a four month period to sample a range of meteorological conditions. Two volcanic eruptions are considered, a 12km eruption of 395 Hekla volcano lasting 24 hours and a 25km eruption of Oraefajokull also lasting 24 hours. To sample different topographical locations 12 different release sites in Europe are considered for the radiological release and in each case the release is assumed to last six hours. For the radiological release scenario total integrated air concentration and total deposition are output after 48 hours. For the volcanic eruption scenario air concentrations of volcanic ash on three vertical levels, deposits of volcanic ash and the total column load of volcanic ash are output every 3 hours. Outputs from the simulations using ensemble meteorology 400 are compared to the outputs from the simulation using deterministic meteorology using the Brier skill score computed for a range of thresholds.

The results showed that on average Brier skill scores were greater than zero for all release locations for the total integrated air concentration and total deposition from the radiological scenarios. This suggests that on average the ensemble dispersion simulation performed better than the deterministic dispersion simulations. Skill scores were greater than zero for all thresholds 405 except the highest threshold for one release site. Skill scores were slightly higher for total deposition than total integrated air concentration. This may be because predictions of precipitation, and therefore predictions of wet deposition, are typically more uncertain than predictions of wind speed and direction giving the ensemble more scope to add value. It also demonstrates that the value of ensemble forecast data depends on the meteorological parameters that have the greatest influence on the output.

Skill scores for the air concentration, deposition and total column load of volcanic ash were greater than zero except at three 410 hours after the start of the eruption where the skill scores for deposition were negative. Zidikheri et al. (2018) used the Brier skill score to compare ensemble simulations of the eruptions of Rinjani and Kelut to satellite observations. Although this study is not directly comparable because model simulations using deterministic meteorology are used in place of observations the Brier skill scores are in good agreement. The skill scores increased with time since the start of the eruption suggesting that the skill of the ensemble increases over time compared to the deterministic simulations. Similar results are observed when the 415 skill of ensemble NWP models is assessed. Examination of individual simulations showed that different skill scores could be obtained for different flight levels so that it was possible for the ensemble to outperform the deterministic in one flight level but not in the neighbouring flight level.



In this study, individual ensemble simulations were compared to analysis simulations to assess whether they outperformed forecast simulations. Using this method uncertainty in the source term and the dispersion model parameterisations is excluded.

The results could, therefore, be viewed as assessing the performance of the NWP ensemble for dispersion applications i.e. the study assesses the NWP parameters of importance to dispersion over scales that are important to dispersion. The data set could also be used to assess whether the dispersion ensemble is *reliable*, i.e. the model predicted probability of an event matches the observed frequency of the same event, and has good *resolution*, i.e. the model is able to distinguish between events which occur with different frequencies. Work to assess this and to increase the range of metrics used to assess the performance of the

ensemble is ongoing and will be addressed in a separate paper.

It would also be useful to determine whether there were certain meteorological regimes where ensemble simulations added more value. Meteorological data is often categorised into weather patterns in order to provide a broad overview of future weather as well as a tool to understand the performance on numerical weather prediction (NWP) models. A comparison of the skill of the ensemble within different weather regimes may indicate the weather patterns where ensemble simulations are

most likely to add value. However, to sample a range of regimes simulations would need to be carried out over a long period of time. In addition, weather patterns are generally applied to broad areas whereas the results of this study have demonstrated that skill can vary over much smaller horizontal and vertical extents. Therefore, it may be more helpful and appropriate to compare ensemble performance to measures of the NWP performance or spread over much small regions.

The aim of this study was to examine the ability of ensemble meteorology to produce more skillful dispersion output than

deterministic meteorology. The study compares ensemble simulations of hypothetical releases to the same simulations carried out with analysis meteorology. Uncertainty in the source term, e.g. release rate, timing, height, and composition, and model parameterisations are ignored. Therefore, the performance of the ensemble seen here may not be reflective of the performance of the ensemble in simulating a real release which would be compared to observations. However, the results do show that on average the ensemble dispersion model outperforms the deterministic model when only meteorology is considered, providing

confidence in the use of ensemble meteorology to provide meteorological uncertainty information to dispersion models. As noted in the introduction, the quantification of uncertainty in dispersion model predictions is important in the decision making process and this study takes the first steps towards demonstrating the value of ensemble dispersion model predictions.

*Code and data availability.*    All the code used within this paper, the dispersion model NAME, the statistical calculation code and the plotting code is available under licence from the Met Office. For access please contact the authors. The meteorological data used to drive the dispersion

model was not generated by this project and is not archived due to the huge volumes involved. Data produced during this work, output from the dispersion model and the results of the statistical calcuations is available via Zenodo at https://doi.org/10.5281/zenodo.4770066 (Leadbetter and Jones, 2021)



*Author contributions.* The project was conceptualised by ARJ and SJL. Formal analysis was carried out by SJL and ARJ. The original draft was written by SJL and SJL, ARJ and MCH contributed to the discussion of the results and the revision of the draft.

*Competing interests.* The authors declare that they have no conflict of interest.

*Acknowledgements.* The authors would like to thank Frances Beckett and Nina Kristiansen (ADAQ, Met Office) for their advice on the volcanic scenario setup and Sarah Millington (ADAQ, Met Office) for her review of a draft of this manuscript.



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





| Radiological Scenario | |
|---|---|
| 48-hour integrated air concentration | 5e4, 1e5, 2e5, 5e5, 1e6, 2e6, 5e6 $Bqs/m^3$ |
| 48-hour accumulated deposition | 5e2, 1e3, 2e3, 5e3, 1e4, 2e4, 5e4 $Bq/m^2$ |
| Volcanic Ash Scenario | |
| Air concentration | 0.2, 0.5, 1.0, 2.0, 5.0, 10.0 $mg/m^3$ |
| Ash column load | 0.2, 0.5, 1.0, 2.0, 5.0, 10.0 $g/m^3$ |
| Accumulated Deposition | 2.0, 5.0, 10.0, 20.0, 50.0, 100.0 $g/m^2$ |

**Table 1.** Thresholds used in the assessment of forecast skill for each of the quantities output from the radiological and volcanic ash scenarios.





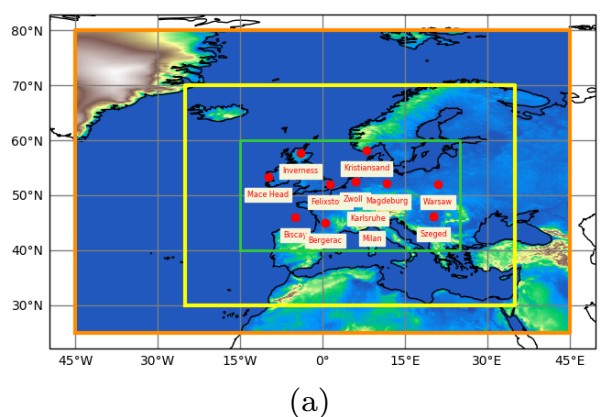

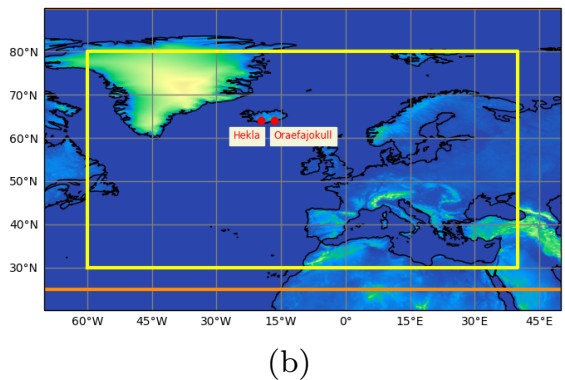

$$(a) \qquad (b)$$

**Figure 1.** Release locations, modelling domain (orange box) and output domain (yellow box) for each scenario. (a) Radiological scenario, (b) volcanic eruption scenario. Note that the modelling domain for the volcanic ash scenario was the whole of the northern hemisphere north of $25^{o}$N.





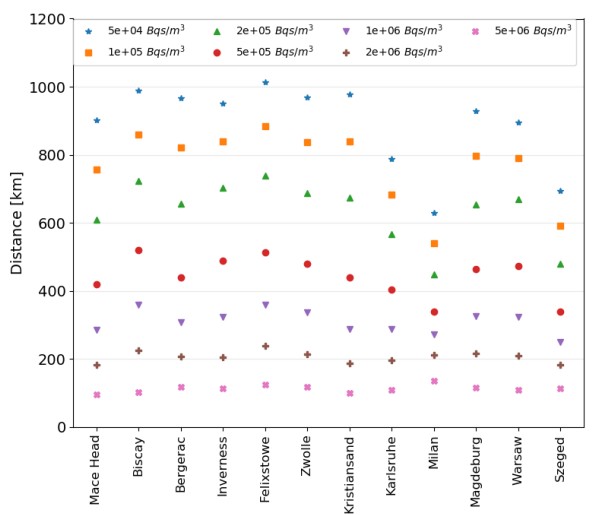

(a) Total Integrated Air Concentration

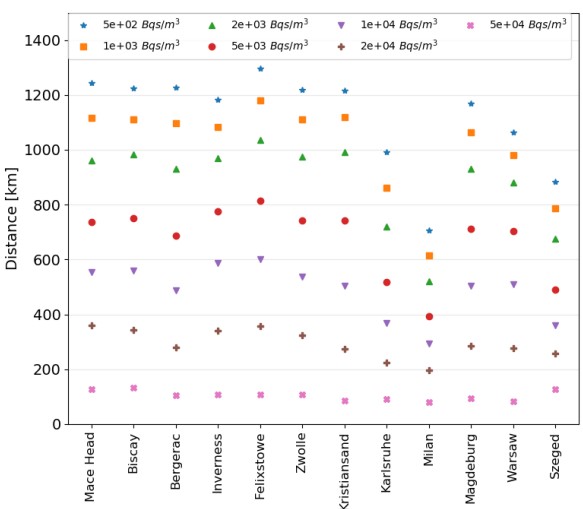

(b) Total Deposition

**Figure 2.** Average maximum distances at which thresholds are exceeeded for (a) total integrated air activity of Cs-137 and (b) total accumulated deposits of Cs-137 for each of the twelve release locations and each of the six thresholds.





(a) FL000-200

(b) FL200-350

(c) FL350-550

**Figure 3.** Average maximum distances at which thresholds are exceeeded for air concentrations on (a)FL000-200, (b) FL200-350 and (c) FL350-550 for each of the six thresholds. The solid line represents average maximum distances from ensemble simulations and the dotted line represents average maximum distances from the analysis simulations of the Hekla scenario.





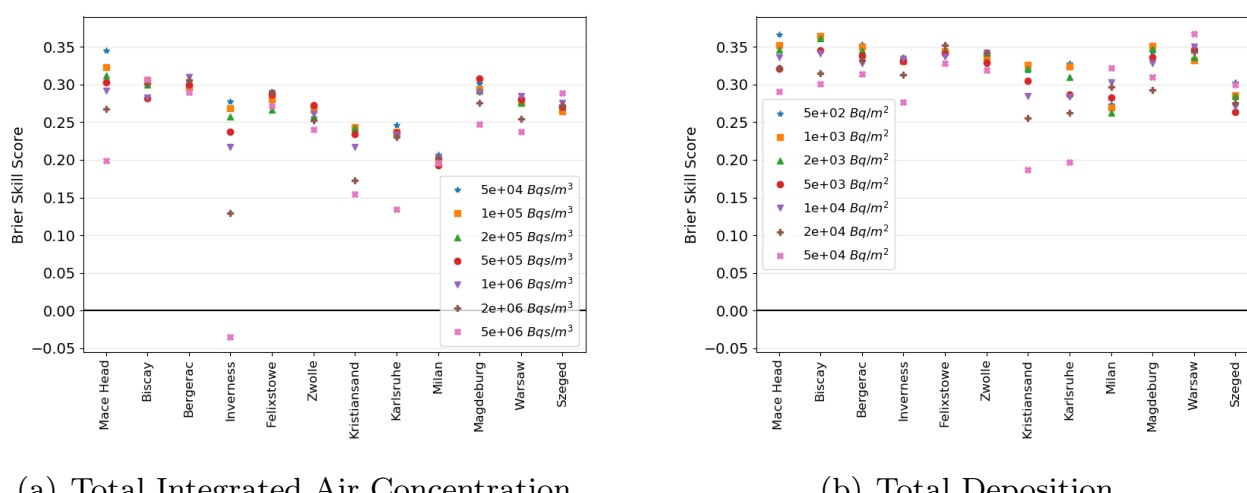

(a) Total Integrated Air Concentration

(b) Total Deposition

**Figure 4.** Average Brier skill score for (a) total integrated air activity of Cs-137 and (b) total accumulated deposits of Cs-137 for each of the twelve release locations and each of the six thresholds.





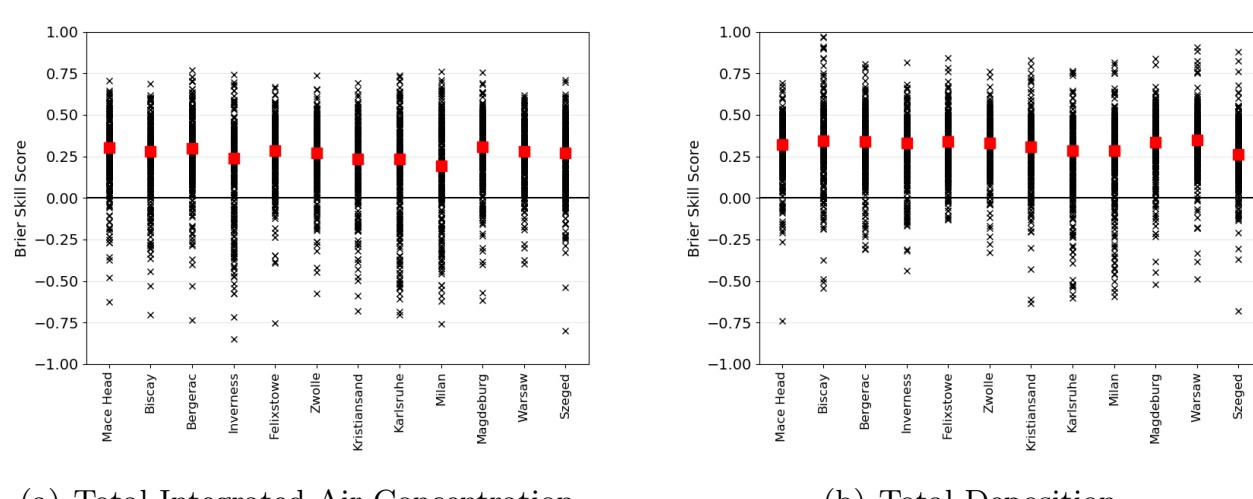

(a) Total Integrated Air Concentration      (b) Total Deposition

**Figure 5.** Adjusted Brier skill score for (a) total integrated air activity of Cs-137 above $50kBqs/m^3$ (b) total accumulated deposits of Cs-137 above $5kBq/m2$ for each of the twelve release locations. Red squares show average Brier skill scores at each location and the black crosses show the adjusted skill scores for each individual simulation.





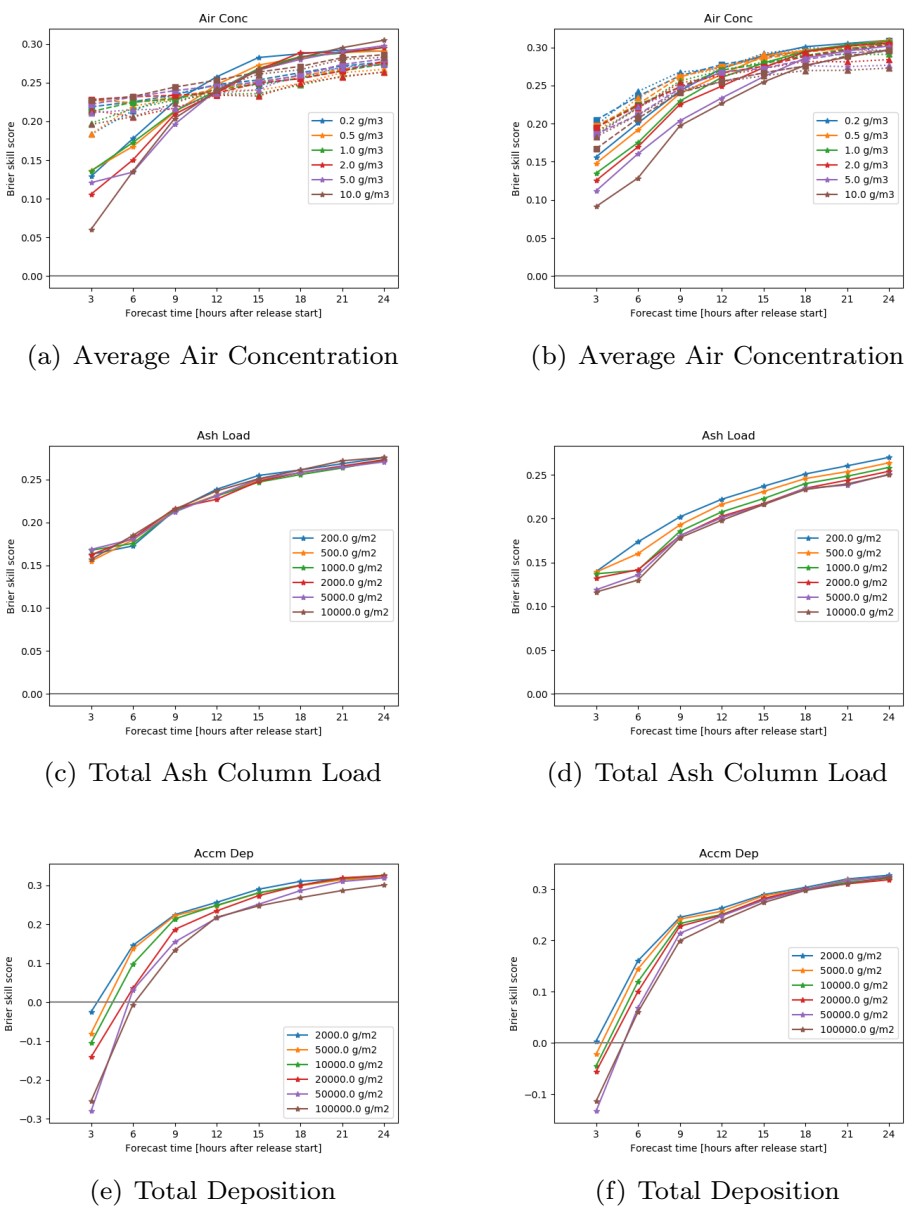

**Figure 6.** Brier skill score against forecast time (hours since the start of the eruption) for (a)(b) ash concentration on three flight levels (solid lines are FL000-200, dashed lines are FL200-350 and dotted lines are FL350-550) (c)(d) ash column load and (e)(f) accumulated deposition for five different thresholds. Left-hand column is for a 12km eruption of Hekla and right-hand column is for a 25km eruption of Oraefajokull.





(a) FL000-200

(b) FL000-200

(c) FL200-350

(d) FL200-350

(e) FL350-550

(f) FL350-550

**Figure 7.** Adjusted Brier skill score against forecast time for ash concentration exceeding $5mg/m^3$ on three flight levels (a)Fl000-200, (b)FL200-350 and (c)FL350-550. Left-hand column is for a 12km eruption of Hekla and right-hand column is for a 25km eruption of Oraefajokull. Each individual run is shown by a black cross and the average Brier skill score is shown as a red square.



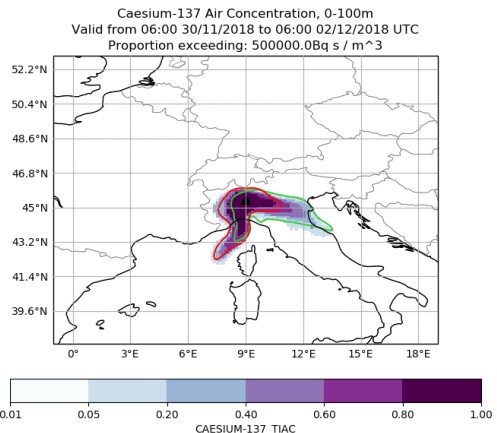

(a) High Brier skill score for air concentration exceeding $500kBqs/m^3$

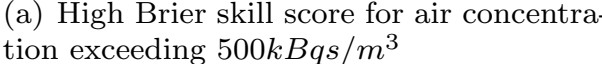

(b) Low Brier skill score for air concentration exceeding $200kBqs/m^2$

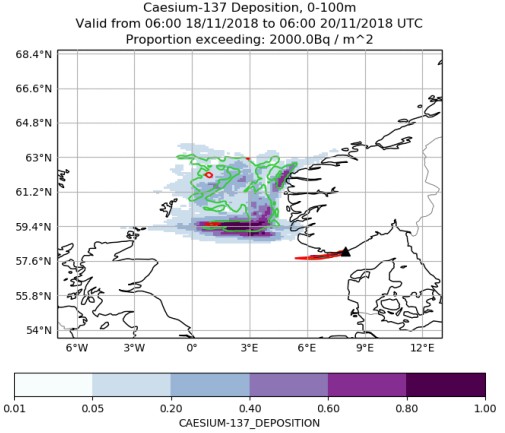

(c) High Brier skill score for deposition exceeding $2kBq/m^2$

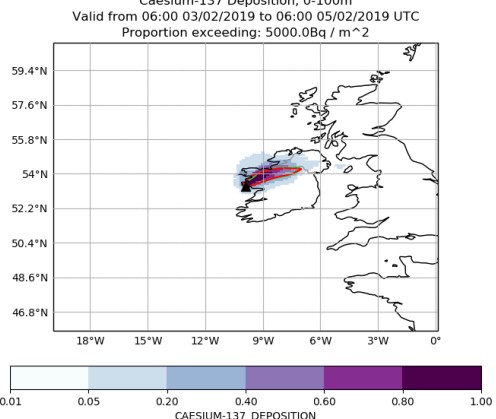

(d) Low Brier skill score for deposition exceeding $5kBq/m^2$

**Figure 8.** Examples of high and low Brier skill score from the radiological scenario. The coloured contours show the probability of the ensemble predictions exceeding the threshold. The area where the threshold is exceeded by the analysis simulation is shown in red and the area where the threshold is exceeded for the deterministic simulation is shown in green.





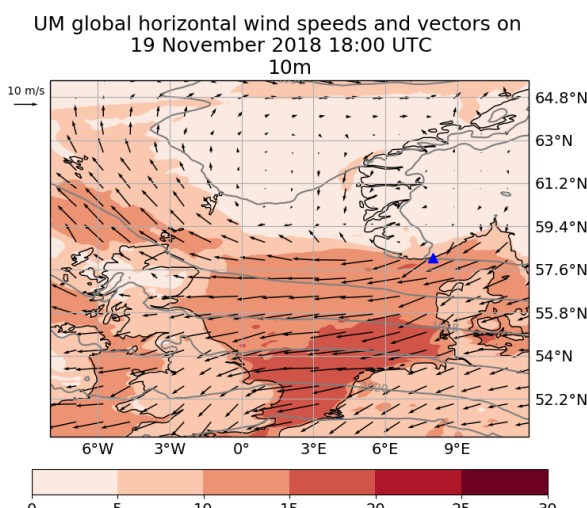

(a) Map of wind vectors, wind speed and sea level pressure

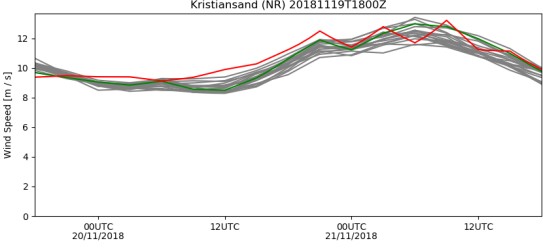

(b) Wind speeds at Kristiansand

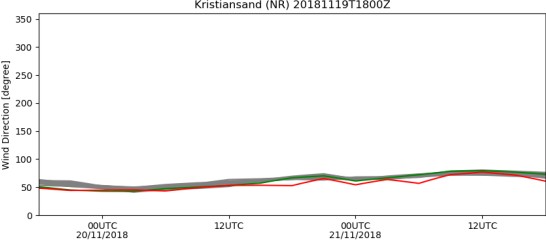

(c) Wind directions at Kristiansand

**Figure 9.** (a) Map of wind vectors (arrows), wind speed (red shading) and sea level pressure (grey lines) from the analysis meteorological data set at 18:00 UTC on 19 November 2018. (b) Wind speed and (c) wind direction at Kristiansand (Norway) from the ensemble forecast (grey lines), deterministic forecast (green line) and analysis meteorology (red line).





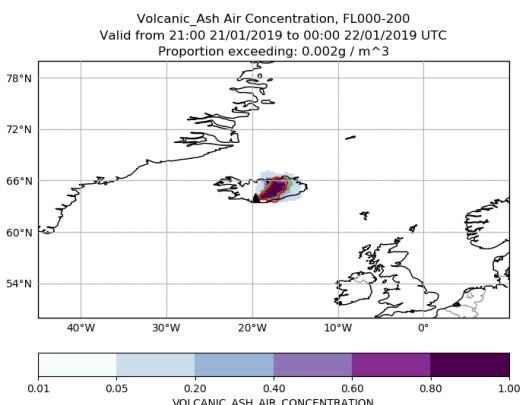

(a) 6 hours after eruption start

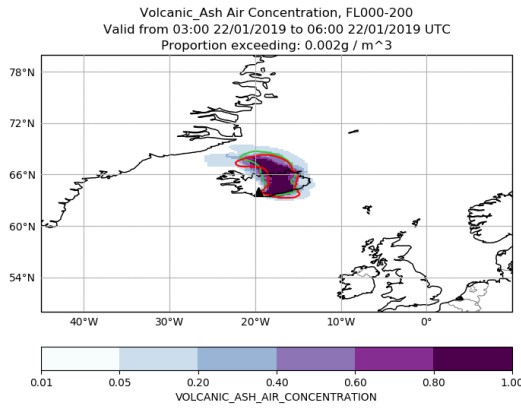

(b) 12 hours after eruption start

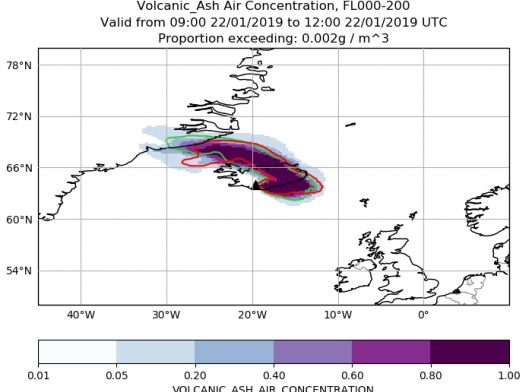

(c) 18 hours after eruption start

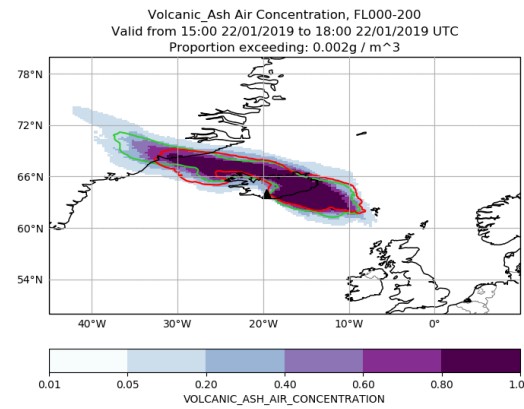

(d) 24 hours after eruption start

**Figure 10.** A volcanic ash simulation of Hekla where the Brier skill score is high for concentrations exceeeding $2mg/m^3$ at FL000-200. The coloured contours show the probability of the ensemble predictions exceeding the threshold in six hour timesteps starting six hours after the eruption. The area where the threshold is exceeded by the analysis simulation is shown in red and the area where the threshold is exceeded for the deterministic simulation is shown in green.

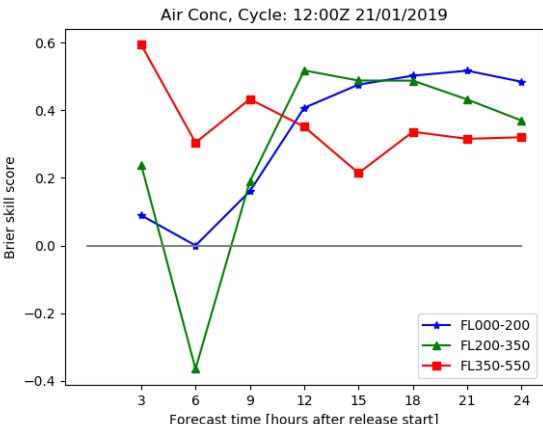

**Figure 11.** Brier skill score changes with time since the start of the eruption for a simulation with a high Brier score. This is for the same case as Figure 10





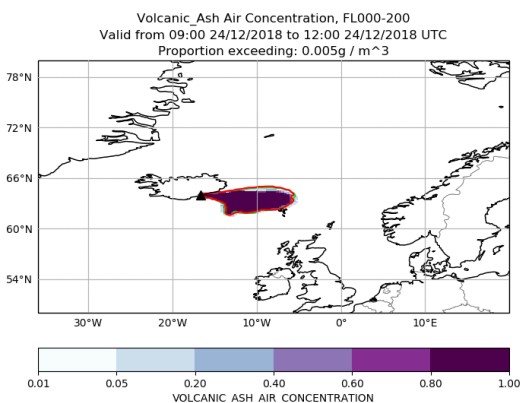

(a) 6 hours after eruption start

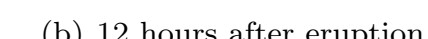

(b) 12 hours after eruption start

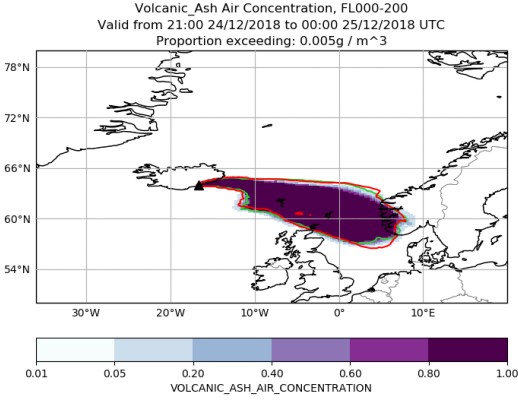

(c) 18 hours after eruption start

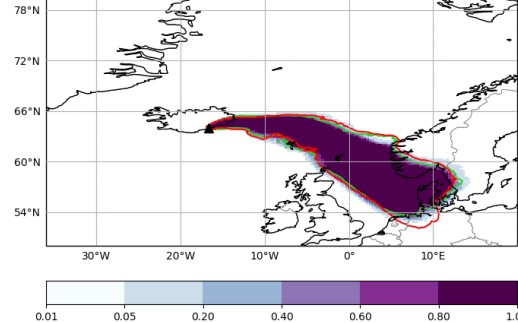

(d) 24 hours after eruption start

**Figure 12.** A volcanic ash simulation of Oraefajokull where the Brier skill score is low for concentrations exceeeding $5mg/m^3$ at FL000-200. The coloured contours show the probability of the ensemble predictions exceeding the threshold in six hour timesteps starting six hours after the eruption. The area where the threshold is exceeded by the analysis simulation is shown in red and the area where the threshold is exceeded for the deterministic simulation is shown in green.




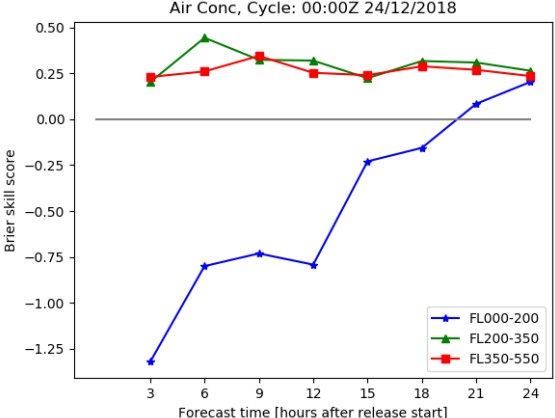

**Figure 13.** Brier skill score changes over time since the start of the eruption for a simulation with a low Brier score. This is for the same case as Figure 12.