# Peer review of "Assessing the value meteorological ensembles add to dispersion modelling using hypothetical releases"

_Atmospheric Chemistry and Physics, 2021_

## Author Comment (AC1)

**Response to Reviewer 1's Comments**

I would like to thank the reviewer for their comprehensive review of the paper. They have highlighted some interesting issues which I have attempted to answer both within the paper and in the response below. The reviewer's comments are shown in italics and my response to their comments is shown in plain text.

*My main comment is related to the period over which the simulations are performed. As explained in L. 153, all the simulation are initialised during a single winter season (2018-2019), but I miss an explanation to why these months were selected for this study and how this choice could potentially influence the results. For example, if persistent weather conditions were present during these four months, this could strongly bias the resulting skill of the ensemble. Also, weather conditions during summer can be very different, therefore I wonder if the statistics would change as well if one were to redo the same experiments for a different period of the year or equally for another winter season. The authors do indicate this potential limitation (L.426-433) and I appreciate that it would require a large amount of extra work to include more months in the study, but I think the paper would benefit from a longer discussion of the potential impact of the selected study period on the robustness of the conclusions.*

The reviewers make a very good point. They are also correct that to extend the study would require many months of work and the data volumes generated are prohibitive. We have, therefore added additional discussion of the limitations of the study both in the methodology (section 2) and the conclusions (section 4) as follows:

In section 2: "To explore a range of meteorological conditions both scenarios were repeated every 12 hours over a period of around 4 months (03/11/2018-28/02/2019 for the radiological scenario and 01/12/2018-31/03/2019 for the volcanic eruption scenario) with each simulation being run on single NWP forecast." has been replaced with "To explore a range of meteorological conditions both scenarios were repeated every 12 hours. Computational constraints restricted the period over which runs could be carried out to 4 months between late autumn 2018 and early spring 2019 so runs were carried out for the period 03/11/2018-28/02/2019 for the radiological scenario and 01/12/2018-31/03/2019 for the volcanic eruption scenario with each simulation being run on single NWP forecast."

In section 4: "Due to computational constraints this study was only able to examine skill scores over a 4-month period from the end of the northern hemisphere Autumn to the beginning of Spring. This was partially mitigated against for the radiological scenario by using a range of release locations. However, further work would need to be carried out to demonstrate that the results hold for the northern hemisphere summer."

*In this study two very different scenarios (near-surface radiological release versus tropospheric/stratospheric volcanic ash eruptions) are explored to examine if the ensemble meteorology produce more skilful dispersion model predictions. Reading through the manuscript, I did miss some discussion that compared the two scenarios in terms of their relative improvements when using the ensemble forecast (i.e. is it more important to consider the ensemble forecasts for boundary layer releases or volcanic eruptions or do they both benefit equally from the ensemble forecast?). In the final paragraph of the paper (L.438-440), the authors mention that on average the ensemble meteorology outperforms the deterministic model. I think the paper would benefit from a short discussion on whether the impact of using the ensemble meteorology is more significant for one of the two scenarios.*

We have added the following paragraph to the conclusions comparing the two scenarios: "Two very different scenarios have been considered, the 48-hour integrated concentrations resulting from a boundary layer release and the time-varying concentrations resulting from a vertical column release over depths of 12 and 25 km. Average Brier skill scores were greater than zero for both scenarios and for all outputs considered suggesting that using ensemble meteorology provides value for a wide range of dispersion scenarios. Brier skill scores tend to be slightly greater for the boundary layer scenario, but further work would be needed to determine whether this was due to the height of the release, the averaging period or the threshold values."

*Finally, I also noticed is that throughout the manuscript (text, legends and figure captions), many of the units are not written in the format as outlined in the ACP submission guidelines (e.g. mg/m3 instead of mg m-3). I have annotated most of them in the technical comments below, but please check carefully throughout the manuscript.*

I've been through the manuscript and corrected the unit formatting.

**Technical corrections/suggestions:**
The reviewer's comments are in a different format.

*L12: '... at those later time steps for deposition than for air concentration.' Based on figure 6 and 7, I am not sure that the differences in BSS between deposition and air concentration at the later timesteps are significant enough to support this statement. Would it be possible to give an estimate of the uncertainty in the average BSS reported in these figures (e.g. standard deviation)?*

We have clarified this statement to indicate that the increase in average BSS over the 24-hour period is greater for deposition than air concentration, since as the reviewers point out this is due to lower average BSS for deposition in earlier time periods rather than higher average BSS for deposition in later time periods.

*L47: comminicate -> communicate*
Done

*L65: need a space between 45 %.*
Done

*L78: The acronym SNAP needs to be spelled out.*
Done

*L84: In the following paragraphs several of these case studies are discussed, but no citation/discussion is presented for the ensemble studies of Eyjafjallajökull and Grimsvötn. I think this should be included.*

Ensembles studies of Grimsvotn are limited but we have included a citation for this case study. We have also included a discussion of the case studies of Eyjafjallajokull.

*L105: used the wrong quotation mark 'dimension'*
Done

*L109: '…use analysis data…' From the text it is not directly clear what analysis model data / meteorology is. I think a short description would be useful, as it is a key part of the analysis presented in this paper. Or alternatively, the authors could refer to L.186.*

The following sentence has been added to clarify the difference between analysis and forecast meteorology: "Here we use analysis meteorology to describe the model meteorological data constructed using a large number of observations to produce a representation of the current state of the atmosphere and forecast meteorology where this atmospheric state is propagated forwards in time."

*L119: '…that NWP ensembles including ensembles may…'. I do not understand this sentence, do you mean: '…that NWP ensembles may…'?*

I have removed "including ensembles" as this is a typographical error and makes the sentence unreadable.

*L130: space needed between 50 m*
Done

*L131: 1 μm should be in roman font.*

Done

*Figure 1: What does the green box represent in panel a? Also, a colour scale for both figures (I assume the colours represent topography) is missing.*

The green box has been removed and a colorbar added.

*L135: '…full 48 hours were output.' Is there a reason to select the 48 hours? In most cases, contaminants will be airborne for longer than the 48 hours. Do we get a different result if we take the output for longer/shorter integration times? I feel some additional argumentation for selecting the 48-hour accumulation period is needed here.*

Contaminants can remain airborne for longer than 48 hours but the time period was limited to 48 hours because this is the time period considered by the UK in the initial response to a radiological accident and also to keep model run times manageable.

*L136: "10km" space between value and unit*
Done

*L138: "12km" and "25km" need space between value and unit*
Done

*L142: "g/hr" should be replaced by g hr-1*
Done

*L143: kg/m3 should be replaced by kg m-3*
Done

*L144: "ash was modelled for 24 hours." Related to the comment for L135, is there a reason why the simulations length is chosen to be 24 hours for the volcanic simulations?*

The simulations were limited to 24 hours, first, because this is the duration of the forecasts VAAC's are required to produce and, second, to keep run times manageable. A note to this effect has been added to the manuscript.

*L.151: "20km" space between value and unit*
Done

*L.153: The period selected for the simulations is one winter season (NDJFM 2018-2019).*
*The aim of the study is to capture a range of meteorological conditions and I am worried*
*that the current selection only captures a limited number of the possible conditions. If a*
*certain weather pattern was dominant during these months (e.g. for the 2018-2019 winter season*
*the NAO was mostly negative), this might have a large impact on the skill of the dispersion model.*
*Furthermore, mass is removed from the model atmosphere by wet and dry deposition, which I*
*assume can be very different in summer especially for the boundary layer releases. All of this will*
*influence the Brier skill score (BSS). I think some discussion regarding this point should be included.*

See response to main comments above.

*L155: remove "the"*
Done

*L161: repetition of definition 'Volcanic Ash Advisory Centre' (L139), so can be omitted*
*here.*
Done

*L176: "20km" space between value and unit*
Done

*L177: "30km" space between value and unit*
Done

*L185: "focus is on the two forecasts at 00 and 12 UTC." I am slightly confused by this*
*sentence, as in the next sentence also the 06 and 18 UTC forecasts are used. Do the*
*authors mean that both for the ensemble and deterministic configurations only the 00 and 12 UTC*
*forecasts are used, but for the analysis meteorology all forecasts are included?*

I've relocated this sentence to the end of the section and expanded to read: In this study dispersion forecasts are initiated only on the forecasts (ensemble and deterministic) at 00 and 12 UTC because the data for NAME is only retrieved for the first 6 hours of the 69-hour update forecasts in order to update the analysis meteorology.

*L206: "…thresholds for the radiological scenario were chosen to reflect typical distances…"*
*There are no references to any literature to indicate that the found values in figure 2 are*
*indeed typical distances for 48 hrs after the occurrence of a radiological dispersion events. I think*
*some information supporting the typical values should be included.*

The reviewers raise a very valid point. This work was carried out following the Horizon2020 CONFIDENCE project (https://portal.iket.kit.edu/CONFIDENCE/index.php?action=confidence&title=objectives) so the setup of the radiological scenario reflects choices made within that project. However, re-reading the CONFIDENCE reports we realised that the distances were based on typical distances at which

deposition thresholds were exceeded rather than air concentration thresholds. The text has been modified to reflect that and to link to the final CONFIDENCE paper.

*L212: exceed -> exceeded*
Done

*L213: "100km" space between value and unit*
Done

*Figure 2: exceeeded -> exceeded. Also need to change the units in the legend to Bsq m-3*
Done

*L214: "650km and 1000km" space between value and unit*
Done

*L215: "700km and 1300km" space between value and unit*
Done

*L222-223: mg/m3 -> mg m-3*
Done

*L225: mg m-3*
Done

*L226: "1 km" space between value and unit and remove italics*
Done

*Figure 3: exceeeded -> exceeded. Also need to change the units in the legend to g m-3*
Done

*L231: "250km" space between value and unit*
Done

*L232: "1600km to 2400km" space between value and unit*
Done

*L238: What could be the impact of the coarser resolution of the ensemble meteorology on the actual dispersion simulated? Would it be possible to downscale the deterministic and analysis meteorology to the same resolution of the ensemble meteorology and run the dispersion simulations with the reduced resolution for one of the cases to test the impact on the results?*

It is possible that the different resolutions of the ensemble meteorology and the deterministic and analysis meteorology has an impact on the results, but we don't believe it is a dominant impact. Most meteorological centres sacrifice resolution for ensemble size, so most ensemble meteorological data are at a lower resolution than their deterministic counterparts. In the event of a real atmospheric dispersion incident, we would use the ensemble and/or deterministic meteorology at their native resolutions and therefore we wished to assess their performance at their native resolutions in this study. We have added a note explaining this below line 239 (original submission).

*Figure 4: units in the legend should be Bqs m-3 instead of Bqs/m3*
Done

*Figure 5: 50kBqs/m3 -> 50 kBqs m-3 and 5kBq/m2 -> 5 kBq m-2. Also, many of the crosses in are overlapping making it hard to understand the density of the crosses. I think a box plot would show the same information more clearly.*

Figure 5 has been replaced with box plots and units changed

*L270: "10 and 30 percent" Are these values for the data shown in figure 5 or is this related to all the different thresholds? Based on figure 4, I would think that the fraction of negative Brier skill scores can be much higher when considering the highest thresholds?*

These are the values for the data shown in figure 5 and we have clarified this at the beginning of line 270.

*Figure 6: Should this be "Averaged Brier skill score…"? The second comment is related to a possible bias in calculating the average Brier skill score (BSS). As mentioned by the authors, the BSS can range from -infinity to 1 (L266), which is why the adjusted BSS was introduced in equation (3). However, in figure 7, several panels show at least one simulation member for the adjusted BSS to be approximately -1. This indicates that the actual BSS was <<-1. If very low BSS occurred for several individual simulations, you could end up with a negative average BSS, even though the rest of the simulations could be near perfect with a score of approximately 1. Therefore, I think it would be useful to report the range of the actual BSS values (or just the minimum) to understand if this could have impacted the average BSS score presented in the paper.*

The reviewer makes and excellent point. The Brier skill score can range from minus infinity to 1 so average Brier skill scores are computed by first computing the average Brier scores for the ensemble and the deterministic meteorology and then using (2) to compute the average Brier skill score. Text to clarify this has been added to the methodology section.

*L281: 'Brier skill scores increase with forecast time,…' How much does the increased number of grid points where we have a plume influence the sensitivity of the Brier skill score? During the initial stages of the simulations, only a small number of points have concentrations above any threshold in all three (ensemble, deterministic and analysis meteorology) simulations, while after e.g. 24 hours the plume has spread over a much larger region (as shown in figure 10). If you misrepresent one grid point of the analysis in the earlier stages for the deterministic and/or the ensemble simulation, will this not have a larger impact on the Brier skill score calculated by equation (2) than the same single grid point error after 24 hours? Is it possible that part of the increase in the Brier skill score with time we see in figure 6 and the reduced range in values in figure 7 for later timesteps is caused by the increased plume size?*

Investigating the impact of the grid size was out of the scope of this project. However, I have plotted the Brier skill score against area of the plume (below). This shows that the spread of Brier skill scores is greater when the area exceeding the threshold is smaller but there is no bias towards negative or positive skill scores for large or small areas.

[Figure]

Figure 1: Brier skill score plotted against area of forecast exceeding the threshold for total integrated air activity of Cs-137 above 50kBqs/m3. The area exceeding the threshold is determined from the analysis plume.

In the text, to help clarify this point we have replaced:

"There are fewer negative scoring runs at later forecast time steps implying that the ensemble is more likely to perform better than the deterministic at later time steps. This is possibly due to the increase in ensemble spread at later time steps."

with "At later forecast time steps there are fewer negative scoring runs and the range of Brier skill scores is narrower. The reduction in negative scoring runs implies that the ensemble is more likely to perform better than the deterministic at later time steps. This is possibly due to the increase in ensemble spread at later time steps. The reduction in the range of the Brier skill scores is likely to be due to the increase in area exceeding the threshold. At early time steps when the plume is narrow the Brier skill score is dominated by a few grid cells and the ensemble tends to be less spread resulting in either a high Brier skill score or a very low Brier skill score. At later time steps the plume is more spread out and the ensemble is more spread so there is a greater range of Brier scores for the different grid cells and the Brier skill score tends to be closer to zero."

*L284: "12km" space between value and unit.*

Done

*Figure 7: Similar to figure 5, I think a box plot would show the same information more clearly. There also seems to be a single simulation in the dataset that seems to perform much better than all the other simulations and shows up on all the panels in this figure. Is this the same simulation for all panels and is there something special about this Simulation?*

Figure 7 has been replaced by a box plot. The single simulation that performed better was erroneous. It was the simulation from the day that there were technical issues with the met data, and it should have been excluded. We have now excluded it from this plot.

*L303: I think this should be 'This resulted in an average Brier skill score of 0.76.'*

As this is a Brier score for a single scenario it is not an average score.

*Figure 8: Please use a different colour for the green or red contour, as it is hard to see the difference. Also replace 'kBqs/m3' -> 'kBqs m-3'.*

The green contour has been replaced by a red dashed contour and the caption amended to reflect this. Units changed

*L305: '200kBqs/m3' -> '200 kBqs m-3'*
Done

*Figure 9: What does the blue triangle in panel a represent? What is the altitude for these wind speed and directions? Are these altitudes at which the contaminants are released in the model?*

The blue triangle represents the release location – a note has been added to caption. The wind speeds and directions are at 10m which is a standard height for meteorological observations and close to the release height of 50m – a note about the height has been added to the caption.

*L315: '2kBq/m2' -> '2 kBq m-2'*
Done

*L334: Why are you using the Brier skill score for the volcanic simulations and not the adjusted Brier skill score like it was done for the radiological scenarios?*

This is a typographical error. I am using the adjusted skill score for the volcanic scenarios and I have modified the text to reflect this.

*L339: 2mg/m3 -> 2 mg m-3*
Done

*Figure 10: Please use a different colour for the green or red contour, as it is hard to see the difference. Also replace 2mg/m3 -> 2 mg m-3.*

The green contour has been replaced by a red dashed contour and the caption amended to reflect this. Changed units

*Figure 12: Please use a different colour for the green or red contour. Also please change 5mg/m3 -> 5 mg m-3.*

The green contour has been replaced by a red dashed contour and the caption amended to reflect this. Changed units

*L365: increase -> increases*
Done

*L433: small -> smaller*
Done

---

## Author Comment (AC2)

**Response to Reviewer 2's Comments**

I would like to thank the reviewer for their comprehensive review of the paper. They have highlighted some interesting issues which I have attempted to answer both within the paper and in the response below. The reviewer's comments are shown in italics and my response to their comments is shown in plain text.

**General comments**

*Line 120, 178 and 420-422. The discussion of configuring ensembles to perform better for certain variables and certain parts of the atmosphere is interesting and would benefit from a lengthier description. In this study, simulations are performed using the MOGREPS-G meteorological ensemble. Has this ensemble been optimised to produce a maximum growth rate of the ensemble spread at a certain forecast lead time? Would differently configured ensembles be more suitable for dispersion applications?*

The configuration of meteorological ensembles and their suitability for dispersion ensembles is a very interesting topic. In the past meteorological ensembles were optimised to produce a maximum growth rate of the ensemble error at a certain forecast lead times but recent work in this field has focussed on ensuring that the ensemble is optimised for all forecast lead times. As far as the authors are aware dispersion studies using ensemble meteorology have focussed on single case studies and single ensemble meteorological data sets (or multi-model ensembles) so have not considered whether differently considered ensembles would be more suitable for dispersion applications. We have added a sentence noting this below line 178.

*Line 331. The authors correctly state that the BSS provides a comparison of the performance of the ensemble relative to the deterministic forecast and does not provide information about the individual performance of the ensemble. Therefore, if the deterministic forecast is accurate the BSS can be negative even if the ensemble forecasts are also representative of the analysis. I would like to see this argument in the introduction section if possible as it's an important point for interpreting these relative skill scores. This is particularly exemplified in figures 12 and 13. By eye the ensemble forecast appears to perform in a very similar manner to the deterministic forecast, but the BSS shows that relatively, this ensemble is worse.*

We have expanded the text mentioning this point in the location where the Brier skill score is first mentioned towards the end of section 2.0.3.

*Line 204, 282, 291 and elsewhere. The Brier Score is calculated for a single output grid square. Does the size of the grid matter? For example, the authors state that the ensemble runs perform better than the deterministic runs at later time steps and hypothesise that this is due to increased ensemble spread at later times. Another reason could be that the plume has spread out more at later times reducing the potential for a double penalty issue. This issue also highlighted in figures 5 and 6, do the negative BSS occur when the plume is narrow, i.e. at the start of the simulations? When calculating BSS at the grid scale small displacements in the plume location can result in large differences compared to the analysis. This occurs particularly when the size of the eddies causing dispersion are large compared to the width of the plume. Would it be possible to show the BSS vs area covered by plume, in an analogous way to fig 7.*

Investigating the impact of the grid size was out of the scope of this project. However, I have plotted the Brier skill score against area of the plume (below). This shows that the spread of Brier skill scores

is greater when the area exceeding the threshold is smaller but there is no bias towards negative or positive skill scores for large or small areas.

[Figure]

Figure 1: Brier skill score plotted against area of forecast exceeding the threshold for total integrated air activity of Cs-137 above 50kBqs/m3. The area exceeding the threshold is determined from the analysis plume.

In the text, to help clarify this point we have replaced:

"There are fewer negative scoring runs at later forecast time steps implying that the ensemble is more likely to perform better than the deterministic at later time steps. This is possibly due to the increase in ensemble spread at later time steps."

with "At later forecast time steps there are fewer negative scoring runs and the range of Brier skill scores is narrower. The reduction in negative scoring runs implies that the ensemble is more likely to perform better than the deterministic at later time steps. This is possibly due to the increase in ensemble spread at later time steps. The reduction in the range of the Brier skill scores is likely to be due to the increase in area exceeding the threshold. At early time steps when the plume is narrow the Brier skill score is dominated by a few grid cells and the ensemble tends to be less spread resulting in either a high Brier skill score or a very low Brier skill score. At later time steps the plume is more spread out and the ensemble is more spread so there is a greater range of Brier scores for the different grid cells and the Brier skill score tends to be closer to zero."

*Finally on line 282; do the authors know why there is a difference in the rate at which the Brier skill score increases with forecast time for different flight levels? One explanation could be that the plume spreads more rapidly at the lower levels due to increased turbulence and remains tightly constrained at upper levels?*

This is an interesting question but unfortunately one which the authors were unable to investigate within the project.

*Although not the aim of this paper, it would be of value in the conclusions to discuss how forecasters/decision makers might make use of ensemble dispersion forecast output.*

The aim of this paper was to examine the value of using meteorological ensembles to provide meteorological uncertainty information to dispersion modelling. An equally important component of the forecasting process is how forecasters and/or decision makers make use of ensemble dispersion forecast information. The authors believe that this part of the process is worthy of a paper (or many

papers) and cannot be covered in a short statement. However, a sentence acknowledging this has been added to the conclusions.

**Specific comments**

*Why is the title posed as a question? The answer is clearly yes, but ensembles add value is what is being addressed here.*

Title modified to: Assessing the value meteorological ensembles add to dispersion modelling using hypothetical releases

*Line 55. The authors refer to the computational expense of running a statistical emulator. In my experience statistical emulators are built precisely because they can mimic the response of a dynamical model but much faster because they only rely on statistical relationships. Perhaps I have misunderstood the meaning of this sentence?*

The reviewer is correct, running a statistical emulator is not computationally expensive. However, a new statistical emulator needs to be constructed for each different dispersion event and therefore the construction of emulators for multiple events is expensive. In addition, to construct emulators for multiple events would require a significant amount of effort. We have corrected this sentence to highlight where the computational expense is.

*Line 75. Not all ensemble systems perturb both the initial model state and the model physics. Therefore 'and' should be 'and/or' in this sentence.*

Sentence modified as suggested

*Line 96. In this section there is reference to the Brier skill score and use of lagged ensembles. These terms should be explained. For example, does the 'most recent ensemble' refer to the lagged ensemble with the shortest lead time?*

I've replaced "They used the Brier skill score to show that a 24-member ensemble performed better than the regional model for both eruptions. However, although the lagged ensemble outperformed the most recent ensemble for Kelut the most recent ensemble performed better for Rinjani."

with "Performance was assessed using the Brier skill score, a skill score that measures the accuracy of probabilistic predictions, to show that a 24-member ensemble performed better than the regional model for both eruptions. \citet{dare:2016} and \citet{zidikheri:2018} also compared the performance of a forecast generated using meteorological data initialized 24 hours earlier than the latest forecast at the start of the eruption and showed that although the ensemble using the older forecast outperformed the forecast using the most recent ensemble for Kelut the forecast using the most recent ensemble performed better for Rinjani."

*Line 99. The term 'dispersion ensembles' is somewhat ambiguous. All Lagrangian model dispersion simulations are ensembles in the sense that they release an ensemble of particles and track their motion. I guess the authors are referring to dispersion simulations run using ensemble of meteorological fields. This is a bit wordy but should be explained in full the first time to avoid ambiguity.*

Replaced: "These studies suggest that for those events that have been examined dispersion ensembles outperform dispersion models run using a single meteorological model."

with: "These studies suggest that for those events that have been examined dispersion models run using ensemble meteorology (hereafter dispersion ensembles) outperform dispersion models run using a single meteorological model."

*Line 151. Here the authors use a 20kmx20km horizontal grid spacing, but earlier (line 136) they use a 10km x 10km grid spacing. Why as a different grid spacing used for the two scenarios?*

The scenarios were set up based on typical grid spacings used within services delivered by the Met Office for volcanic ash forecasting and radiological dispersion forecasting. An increase in resolution was applied to both scenarios to reflect likely future increases in resolution of both services.

*Line 213. Do the authors have a reason or hypothesis for why the highest threshold is exceeded around 100km from the release location for all the release locations?*

This work was carried out following the Horizon2020 CONFIDENCE project (https://portal.iket.kit.edu/CONFIDENCE/index.php?action=confidence&title=objectives) so the setup of the radiological scenario reflects choices made within that project. So, the threshold exceedance at around 100km was chosen based on the distance thresholds were exceeded in the modelling carried out in that project. However, re-reading the CONFIDENCE reports we realised that the distances were based on typical distances at which deposition thresholds were exceeded rather than air concentration thresholds. The text has been modified to reflect that and to link to the final CONFIDENCE paper.

*Line 239. The analysis and deterministic met have the same grid spacing while the ensemble met has coarser grid spacing. Does this impact the results? If so, why not coarse grain the analysis and deterministic met to the same grid spacing as the ensemble met?*

It is possible that the different resolutions of the ensemble meteorology and the deterministic and analysis meteorology has an impact on the results, but we don't believe it is a dominant impact. Most meteorological centres sacrifice resolution for ensemble size, so most ensemble meteorological data are at a lower resolution than their deterministic counterparts. In the event of a real atmospheric dispersion incident, we would use the ensemble and/or deterministic meteorology at their native resolutions and therefore we wished to assess their performance at their native resolutions in this study. We have added a note explaining this below line 239 (original submission).

*Lines 358-400. These two paragraphs are a repetition of the methodology and are not conclusions. Therefore, it is not appropriate for them to be in the conclusions section.*

This section has been re-titled as summary and conclusions.

*Table 1. What time period are the accumulations over?*

The volcanic ash is accumulated from the start of the eruption up to the forecast time so for the 3-hour forecast time it is the total accumulated over 3 hours, for the 12-hour forecast time it is the total accumulated over 12 hours.

*Figure 2, 3 and 4. Are the averages over all releases?*

Yes

*Figure 3. Why is there a larger spread in the average maximum distances for the different concentration thresholds in the lowest layer (FL000-200) compared to the higher layers? Is this due to deposition of particles to the surface? If the particles did not deposit to the surface does this difference in spread decrease?*

Investigating the spread in average maximum distances for the different concentration thresholds was out of the scope of this project.

**Typographical errors**

Both errors below have been corrected.

*Line 146. There is an extra space before 800m.*

*Line 308. 'at and' should be 'and'*

---

## Author Comment (AC3)

**Response to Reviewer 3's Comments**

We would like to thank the reviewer for their comprehensive review of the paper. They have highlighted some interesting issues which we have attempted to answer both within the paper and in the response below. The reviewer's comments are shown in italics and our response to their comments is shown in plain text.

**Specific comments**

*In the Introduction section, reference is given to earlier work on the use of ensemble techniques for atmospheric dispersion modelling. However, there is no reference to the research carried out by the Nordic countries on this issue in a number of projects, cf. e.g. Sørensen et al. (2020), see below:*

*Sørensen, J.H., Bartnicki, J., Blixt Buhr, A.M., Feddersen, H., Hoe, S.C., Israelson, C., Klein, H., Lauritzen, B., Lindgren, J., Schönfeldt, F., Sigg, R. Uncertainties in atmospheric dispersion modelling during nuclear accidents. J. Environ. Radioact. 222 (2020) 1-10. https://doi.org/10.1016/j.jenvrad.2020.106356*

I've now added additional references to earlier work on ensemble techniques within the introduction section. These include comment on improving ensemble methodologies leading to lower computational requirements and citations for ensemble studies of Grimsvotn and Eyjafjallajokull.

*For the selected scenarios, four months of meteorological data (mainly winter 2018-2019) are selected. However, no reasoning for this choice is given. A whole year would seem more appropriate. Please comment on this.*

The reviewers make a very good point. However, to extend the study would require many months of work and the data volumes generated are prohibitive. We have, therefore added additional discussion of the limitations of the study both in the methodology (section 2) and the conclusions (section 4) as follows:

In section 2: "To explore a range of meteorological conditions both scenarios were repeated every 12 hours over a period of around 4 months (03/11/2018-28/02/2019 for the radiological scenario and 01/12/2018-31/03/2019 for the volcanic eruption scenario) with each simulation being run on single NWP forecast." has been replaced with "To explore a range of meteorological conditions both scenarios were repeated every 12 hours. Computational constraints restricted the period over which runs could be carried out to 4 months between late autumn 2018 and early spring 2019 so runs were carried out for the period 03/11/2018-28/02/2019 for the radiological scenario and 01/12/2018-31/03/2019 for the volcanic eruption scenario with each simulation being run on single NWP forecast."

In section 4: "Due to computational constraints this study was only able to examine skill scores over a 4-month period from the end of the northern hemisphere Autumn to the beginning of Spring. This was partially mitigated against for the radiological scenario by using a range of release locations. However, further work would need to be carried out to demonstrate that the results hold for the northern hemisphere summer."

**Technical corrections**

*Line 47: Replace comminicate with communicate.*

Done

*Line 119-120: Incomprehensible sentence: (…) NWP ensembles including ensembles (…)*

Removed "including ensembles!

*Line 268: The range [−1, 0) should be corrected to ]−1, 0[ or ]−1, 0).*

In the current format we agree that it is unclear whether the lower end of the range is open-ended or closed. We have restated equation 3 to make it clearer that the lower end of the range is closed and that "[" is correct.

*Line 274: Replace air concentration with integrated air concentration.*

Done